# Experimental Investigation of Cohesion between UHPC and NSC Utilising Interface Protrusions

**DOI:** 10.3390/ma15196537

**Published:** 2022-09-21

**Authors:** Pavel Horák, Šárka Pešková, Marcel Jogl, Radoslav Sovják, Petr Vítek

**Affiliations:** 1Faculty of Civil Engineering, Czech Technical University in Prague, Thákurova 7, 16629 Prague, Czech Republic; 2HOCHTIEF CZ a.s., Plzeňská 3217, 15000 Prague, Czech Republic

**Keywords:** concrete, UHPC, interface, interaction, shear, cohesion, button foil, protrusions

## Abstract

The interaction of ultrahigh-performance concrete (UHPC) and normal-strength concrete (NSC) is one of the main issues for strengthening conventional concrete structures or other applications where NSC and UHPC are interrelated. UHPC stands out for its strength and durability, while NSC is significantly inexpensive and easier to work with. Efficiently designed structures can exploit the advantages of both mixtures. At the interface of these materials in newly designed structures, the formwork can be modified at the interface to give the concrete surface sufficient roughness and thus cohesion as required. This improves both the tensile and shear strength of the contact resulting in the enhanced capacity of the composite structure. In this study, a button foil was inserted into the formwork for the UHPC and then a part of NSC was made. The shear strength of the interface without any stress component in the transverse direction was measured on small-scale samples. It was to justify the possibility of the use of this interface in real constructions such as beams and columns. The main objective of further research is to design a composite beam using a UHPC shell as formwork for NSC and protrusions at the interface. It is expected that the U-shaped shell made of the UHPC could significantly contribute to the load-bearing capacity of the resulting composite NSC–UHPC structure and also to its enhanced durability. In addition, if the NSC is enclosed in a shell of UHPC, it can be made from various secondary materials, therefore it can decrease cement consumption by more than 50%.

## 1. Introduction

UHPC is a relatively young, but rapidly expanding material that offers an extensive range of applications. When placed in the formwork, it behaves similarly to self-compacting concrete. However, there is a risk of segregation of the dispersed reinforcement. Due to the fine microstructure, the surface appearance is usually better than that of conventional concrete. Moreover, the resistance of the UHPC surface to mechanical damage, abrasion and chemical resistance is much better compared to that of NSC [1]. The service life of structures made from UHPC is expected to be 200 years or more. On the other hand, there is a risk of brittle failure, not only under extreme load, due to a much smaller cross-section. Therefore, fibres in the form of disperse reinforcement are an important part of the mixture, resulting in a higher ductility and energy absorption capacity. The mixture with dispersed fibres also provides better resistance to cyclic loading [2].

The strength of UHPC can be increased not only by adding dispersed reinforcement, e.g., steel fibers, but also by adding silica fume or by steam curing [3]. Silica fume fills the empty pores in the mixture and helps to distribute the cement grains homogeneously [1]. This also leads to a higher percentage of cement hydration. Adding the optimum amount of nanosilica and silica fume powder can increase the tensile and compressive strength of concrete by more than 100% [4]. Effective steam curing increases strength, modulus of elasticity, reduces creep, eliminates drying shrinkage, reduces permeability and increases resistance [5].

Due to the fact that UHPC achieves a high load-bearing capacity even with significantly lower dead weight of elements than NSC and has high requirements for concreting, it seems advantageous to use this material for precast elements. UHPC can be used alone in the structures, for example as segments of footbridge [6] or in combination with other materials (especially other types of concrete). These substructures can be beams with the subsequent addition of an NSC layer [7] or bridge ledge [8]. UHPC can also interact with a higher strength material in the structure as in the case of [9]. The subject of this research was a high strength steel–UHPC composite beam, where UHPC formed the top plate of the beam, and the bottom part was made of an H-shaped steel beam.

The area of contact between the UHPC and the conventional normal strength concrete is significant. In existing structures, a layer of UHPC can be added to improve and strengthen the structure [1]. In new structures, the reverse procedure can be advantageously applied, i.e., parts of the UHPC can be supplemented with NSC. However, in many cases, the UHPC serves mainly as a lost formwork and the interaction is no longer verified. Concrete filling can even be completely isolated from the surrounding environment (filling concrete), e.g., in U-shaped beams [1] or columns [10]. Then, the input materials of this fill can be supplemented with secondary materials such as fly ash or slag, among others.

The parameters of the materials and the area of contact determine the interface’s load-bearing capacity. One of the possibilities to improve this area is to create textures. The basic textures are smooth (as-cast), deep-grooved, longitudinal-roughened, transverse-roughened and indented surfaces, which were compared in the study [11] for the NSC–NSC interface. The indented profile had 5 mm high linear outcrops similar to the basic trapezoidal profile. For specimens with no stress component to prestressed concrete parts together, the best shear strength of 3.78 MPa was achieved by the transverse-roughened profile. The lowest shear strength of 0.61 MPa was recorded for the smooth surface.

The shear strength is also affected by the age of the mixtures, the strength of the NSC, the method of concrete treatment and the composition of the mixtures [12]. In this case, the NSC section was concreted first and afterwards the UHPC section was made. This research showed that the quality of both materials and the surface treatment significantly impacted on the shear strength of the interface. At the smooth UHPC–NSC interface, a shear strength of 4.42 MPa was measured, while for the roughened surface, the shear strength reached a value of 6.55 MPa. Thus, a suitable surface treatment increased the shear strength by almost 50%.

The approach for determining the shear strength at the interface between concrete layers or between concrete and UHPC layers is defined in standards such as Eurocode 2, ACI 318 or FIB Model Code 2010. These standards give specific recommendations for calculating the shear strength at the interface according to the surface roughness. The result is the friction and cohesion coefficient of Mohr–Coulomb failure criterion, which defines the shear strength according to the applied normal force described for the UHPC–NSC interface in [13]. The shear surface is inclined and the applied force has a normal and friction component depending on the inclination angle.

The next method of determining shear strength is called the push-off test. The normal force is directly applied to the specimen and the shear strength is measured. This method has been used in many studies [11,14,15,16].

### 1.1. Technical Background

The interfaces can be the weakest element of the whole load-bearing part in the structures made from different cementitious materials. They have to be well-adjusted, especially if they directly affect the load bearing capacity of the structure. This is why numerical models should focus on the best possible interface approximation. It is assumed that the UHPC part is delivered to the site in precast form and subsequently filled with NSC in construction and a duct can also be made there for additional prestressing.

There are several approaches to modelling this type of contact for a finite element method. The simplest is to prescribe a strong bond assuming that the crack occurs outside the interface of the materials [17,18]. In the case of [19], the cohesion was modelled with only a shear strength of 1.9 MPa and a friction coefficient of 1.5. However, these values were not experimentally verified. The ideal elastic–plastic model was used in the case of [20]. However, the nondecreasing shear force at failure did not correspond to the reality [7].

The damage model used in [21,22] is a better representation of the material contact. However, some of the models in software ATENA, DIANA or Abaqus become mathematically ill-conditioned in the contact damage region and the simulation result then lacks a physical meaning [23,24] or the computational time increases significantly. It depends on each structure and the required accuracy of how the contact can be modelled.

An experiment simulating a UHPC beam with an overlayer of NSC [7] described the interface shear strength based on experimental data and numerical modelling. A variation of the shear strength with friction coefficient was used here due to the complexity of the model. The shear strength thus depended on the perpendicular load. The model was calibrated on specimens with different shear surface inclination angles depending on the applied force at failure. As can be seen from that paper, the dependence of shear strength and normal force was almost linear.

A common example of shear or tensile failure of UHPC and NSC cohesion tests is a collapse in the noninterface part in NSC [13,17,25]. However, this is not constant and the location of the crack depends on both the experimental setup and material properties [26]. In real structures, the human factor and the element of coincidence [7] could not be neglected. Thus, whether the models can predict collapse outside the interface should also be verified by experiments depending on the mixture compositions and surface treatment of the UHPC at the interface.

An important and still understudied area is cyclic load-bearing capacity and the long-term interaction between UHPC and NSC, as the first UHPC structures were built around 2000. Only a minimum of studies deals with these issues.

### 1.2. Research Significance

The aim of this study was to verify the interaction between UHPC and NSC. It further served as the main basis for the design and experimental verification of interaction in real structures such as beams and columns.

The contact area contains protrusions to increase cohesion. The future goal of the research is to develop a new composite beam consisting of a filigree prefabricate of a suitable shape from UHPC, with the subsequent implementation of selected secondary materials. The composite material will be of better quality and will reduce economic costs associated with the operation, durability, frequency of repairs and inspections of bridge structures. The project itself focuses on the application of new technical and technological solutions for transport structures to increase the service life of structures using recycled materials and streamline the maintenance of bridge structures, indirectly leading to reducing the negative impact on the environment due to material savings and CO${}_{2}$ reduction associated with cement production. The solution supports the main goal of the Sustainable Transport program and focuses on maintaining the competitiveness of transport in order to permanently increase the efficiency of the transport system. The solution targets bridges with fields over operated highways and railroads. The other benefits of the project are:Shortening the construction time;A quality, durable construction surface;Restrictions on formwork elements on site;Restrictions on handling temporary structures;Increasing the usability of secondary raw materials and promoting raw material self-sufficiency.

## 2. Materials and Methods

### 2.1. Materials

The first material used was UHPC. Ready-mixed concrete was bought as a product, and its detailed properties are the know-how of the company [27]. The cube strength of this material usually ranges from 125 to 150 MPa. The processing time of this mixture can be extended up to 3 h. The selected mixture contained approximately 2% by volume of crimped steel fibres of length 17 mm and diameter of 0.3 mm. The D${}_{max}$ of the aggregate was 8 mm.

Another material used in this study was normal-strength concrete (NCS). This material was mixed in the laboratories of CTU in Prague. Efforts were made to replace some of the cement with another more environmentally friendly binder. One suitable binder was developed in the CTU in Prague, resulting in the ecological fluidized bed combustion-based ternary binder (FBC-TB) [13,28,29,30]. The binder is suitable for the production of concrete, mortar, prefabricated products and vibropressed products. It is also suitable to use as an admixture in concrete, in the construction layers of roads and to improve soil properties. FBC-TB is a composition of hydrated products (presence of amorphous C-A-S-H phase) similar to Roman concrete [30]. Materials with a zero carbon footprint are used for its production and therefore result in a very significant reduction of CO${}_{2}$ production. One ton of binder used saves 0.72 t of CO${}_{2}$ compared to using conventional cement [28].

FBC-TB is currently used in the construction of several structures, e.g., the production of shotcrete for the construction of the Prague metro line D [29], as well as for the construction of paved roads, cycle paths, road panels and prefabricated parts for retaining walls. It is also used for some special applications.

In this study, a total of 3 mixtures were used. These were UHPC, normal-strength concrete NSC1, which served as a comparison mixture, and the concrete with partial replacement of cement by FBC-TB further termed as NSC2. The composition of both mixtures can be seen from Table 1. The shear strength of UHPC–conventional concrete and UHPC–FBC-TB concrete interfaces was tested. One of the main aims was to verify the consequence of the use of FBC-TB in concrete on the shear strength at the interface. The mechanical properties of each mixture were clarified, as can be seen in Table 2.

The elastic modulus, compressive strength on cylinders and cubes and simple tensile strength on dogbones were tested on the specimens. The modulus of elasticity was measured on the cylinders with a diameter of 150 mm and a height of 300 mm. The tensile strength was tested on specimens where a tensile crack occurred in a part of a rectangular cross-section of 50 × 100 mm according to [31]. In this specimen, according to [32], a size effect could be significant for UHPC because the specimen thickness was only 50 mm in the middle part where the crack occurred. Dispersed reinforcement in small specimens tends to be parallel to the formwork and the surface planes, and in this narrow part, fibres would be more in the longitudinal direction. The angle measured between the dispersed reinforcement and the specimen axis significantly affects measured tensile strength as shown in Figure 1. Therefore, only mixtures without dispersed reinforcement were tested by this method.

Furthermore, the tensile flexural strength of each material was tested using both the three-point bending method and the four-point bending method. The three-point bending was tested on 100 × 100 × 400 mm size beams and the four-point bending on the same size beams. The size effect mentioned above on these specimens was clearly present and may have increased the measured strength by a few up to dozens of percent [32].

The results from each type of examination are listed in Table 2. The compressive strengths on 150 × 150 × 150 mm cubes are denoted as C-Cube, the compressive strengths on 300 mm high and 150 mm diameter cylinders as C-Cylinder. The simple tensile strength is denoted as T-Simple and flexural tensile strengths as T-Bend. The number depends on whether it is a three-point or four-point bending test. The last item in this table is the modulus of elasticity E.

The abbreviation NSC1 stands for conventional concrete with the designation C40/50. In NSC2, half of the amount of cement was replaced by FBC-TB. This means that about 180 kg less cement was used per 1 m${}^{3}$, which also reduced the carbon footprint of the production of this concrete. Compared to NSC2, NSC1 had a significantly higher compressive strength both on cubes and cylinders, on average by more than 35%. This ratio remained approximately the same even for simple tensile tests. One specimen made from NSC2 failed at a significantly lower load. No cavity or damage was found in the part where the failure occurred. This could have been, for example, preinduced damage.

### 2.2. Methods

Box and sandwich specimens were created to determine the shear strength of the interface. A button foil was inserted into the inner part of the formwork for UHPC to guarantee the roughness of the interface. Buttons made from the UHPC protruded into the NSC. The geometry of these protrusions can be seen in Figure 2. Button foil is commonly used to separate drainage fill and building walls at below-grade levels. This ensures the ventilation of the building walls and reduces the amount of moisture on the surface of the walls.

Both types of specimens were made in a total of six pieces. Three pieces of UHPC were filled with conventional concrete, and three pieces were filled with concrete where cement was partially replaced by FBC-TB. The height of the outcrops of the button foil was 8 mm, and the shape was a conical frustum with a larger diameter of 16 mm at the base and smaller at the top of 8 mm. These buttons were arranged in a square grid at an axial distance of 23 mm. These protrusions can be seen on Figure 2. The total density of the outcrops was about 1890 buttons per 1 m${}^{2}$. The foil was placed on the formwork so that the squares were rotated by 45${}^{\circ }$.

The sandwich specimens were composed of two 100 mm thick UHPC slabs. The space of 100 mm between them was filled with NSC according to [1]. The specimen was concreted in the horizontal position, which can be seen in Figure 3 and Figure 4. In order to prevent the plates from separating without achieving the shear strength of the interface, a threaded rod was inserted in the middle part. This kept the two parts of the UHPC together with no significant additional load. The nuts were not further tightened after concreting. The required distance was maintained by the embedded extruded Styrofoam. The part below the Styrofoam was also filled with NSC to prevent the UHPC elements from rotating significantly under load. The upper part (the closer one in Figure 4) protruded from the UHPC so that it could be loaded, as it can be seen in Figure 5. However, it was ensured that the filling concrete did not create a rim at the base of the protruding part that would fictitiously increase the force required to achieve the shear strength of the specimen. In this case, the area of the interface was 2×200×200 mm.

The second type of specimens for interface shear strength verification were box specimens, see Figure 6. All four inner sides of UHPC in the area of further contact with NSC were made with protrusions. Concreting was done vertically with the top protruding of NSC for an easier layout of the destruction experiment. The UHPC was reinforced with five reinforcing stirrups with a diameter of 6 mm to avoid brittle tensile failure of this outer part during destruction. Styrofoam of 100 mm height was placed at the bottom part of the box to enable vertical movements of the inner NSC part. As with the sandwich specimen, the footing of the protruding portion of the infill concrete was modified so as not to affect the measured interface shear strength. The interface area was 4×200×200 mm.

In the closer part of Figure 7, there are three specimens made from UHPC just filled with NSC1. Foil is placed over the top of the samples to prevent unwanted drying of the concrete. Behind these specimens, there are 3 other specimens prepared to be filled with NSC2.

## 3. Results and Discussion

Table 3 and Table 4 show the shear strengths obtained for both types of interfaces on both types of specimens. The measured values represent the minimum shear strength reached with no stress component in the transverse direction at the interface in real structures. The sandwich samples were subjected to compression, as can be seen in Figure 8. As the Styrofoam was compressed, the crack in the upper part was opening. The test was stopped after the shear force had stabilized depending on the deformation or when the threaded rod had affected the measured value. For the mixture NSC2, the measured shear force was significantly lower than that for the mixture NSC1. An average shear strength of 3.85 MPa was measured for the NSC1 mixture. However, the average value comprised only two values as the third specimen failed. Again, the failure surface showed no evidence of cavern formation or a higher incidence of air bubbles.

No significant damage was visible on the box-type sample after deformation. The inner part had been pushed inwards while only small cracks occurred, as can be seen in Figure 9. These cracks were caused after separating the protrusions when the shear strength of the interface was reached. The cracks grew larger with increasing deformation until they reached the size shown in the figure. The shear strength measured on the box specimens was several times higher than on the second type of specimens. At the UHPC–NSC2 interface, a strength of up to 8.6 MPa in shear was measured on this type of specimen. Shear strength measured on the interface UHPC–NSC1 was comparable to the strength measured on UHPC–NSC2.

The results of the UHPC and NSC interaction tests showed that the crack occurred mainly in the plane from which the buttons protruded. The NSC and UHPC separated at this part, and the shear of the buttons occurred. A part of the dispersed reinforcement was broken, and a part was separated from the matrix of the UHPC. This can be seen in Figure 10 on a sandwich-type of specimen and the UHPC–NSC2 interface. There is a hole for the threaded rod in the middle part. The buttons in this part are not damaged because they were not in contact with the NSC but with the Styrofoam. The coherence with the rest of the UHPC plate is still visible on the preserved sheared buttons. In the lower left part of the figure, there is an area where the crack occurred, mainly in the NSC part.

Figure 11 shows the evolution of the deformation depending on the applied stress. For clarity, each type of sample and material interface is represented by only one typical line. Designation S means a sandwich sample and K means a box sample. Each sample has a number: 1, 2 and 3 are for those filled with NSC1 and 4, 5 and 6 with NSC2.

The red and yellow lines show the strain evolution on the box sample, and the blue and purple lines on the sandwich sample. In the experiment, a wooden plate was placed between the top part of the sample and the metal surface of the hydraulic loading machine. This ensured a more uniform force transfer over the whole surface and did not result in crushing protrusions or concrete failure in the top part.

The use of this spreader plate explains the difference between the slope of the curves in the diagrams. After the plate was compressed and the plastic deformation of the wood had occurred, lines settled into a linear progression. The slope of all curves during the initial deformation is almost the same when the effect of the elastic wood deformation is neglected.

The box specimens reached a strength of almost 8 MPa. After damage, the stress dropped by approximately 2 MPa. There was almost no difference between the UHPC–NSC1 and UHPC–NSC2 interfaces. These specimens also showed a very high residual strength, which remained above 4 MPa even after a 15 mm slip. This indicated that the interface could perform its function even after a shear or shrinkage crack had formed.

A wooden plate that had already been compressed several times was placed on top of the second type of specimen, so its deformation was hardly reflected in the graph. At a load of approximately 1.5 MPa in both types of sandwich samples, there was a slight increase in the deformation. Most likely, this was not a deformation of the UHPC–NSC interface. After the increase in deformation, the curves showed an increasing trend up to a value of 2.4 and almost 4 MPa. After reaching this value, there was a significant loss of strength and a rapid increase in strain. The residual strength was below 1 MPa. This value would be noticeably higher if the transverse deformation was prevented.

The measured residual strength after the first crack occurred also determined the characteristic shear or tensile strength of the interface. This approach guaranteed the required ductility and prevent a brittle failure of the structure [1].

The results obtained for the sandwich samples made from UHPC and NSC1 were confirmed by a similar experiment [33] aimed at comparing different sizes of linear protrusions in the contact area. The height of the protrusions that achieved the best shear strength was 10 mm, and the shear strength obtained was 3.73 MPa. This value was very close to the shear strength of the NSC. The shear strength of the interface with no protrusions was less than 1 MPa. This showed very well the importance of the protrusions or textures in the area of the interface.

Figure 12 compares the normalized shear pressure from Figure 11 as a function of strain. It was normalized by the average value of the compressive strength of NSC measured on cubes according to Table 2. An almost identical normalized shear strength was reached for the sandwich-type specimens. The residual strength was higher in the case of the UHPC–NSC2 interface. The normalized shear strength of the UHPC–NSC2 interface for the box-type specimens was approximately 25% higher than that of the UHPC–NSC1 interface. This showed that the strength of NSC affected mainly the shear strength of the sandwich specimen interface.

Creating protrusions using button foil is a more straightforward and more time-efficient method than creating linear protrusions. The measured shear strength was almost identical to that measured at the interface of UHPC and NSC1. However, the NSC1 mixture achieved a higher cubic strength. The NSC2 mixture, which had a similar cubic strength, achieved a 35% lower shear strength at the interface with the UHPC.

When compared with the same experiment as mentioned above [12], exactly the same value of the shear strength at the UHPC–NSC1 interface was obtained for the sandwich type specimen as for the shear strength on the smooth surface between UHPC and NSC. However, the reverse process was applied here, i.e., a part of UHPC was concreted later than the part of NSC. This significantly increased the shear strength at the interface [1].

The box-type specimens with both UHPC–NSC1 and UHPC–NSC2 interfaces achieved higher shear strengths than those achieved in the study [12] by any modification of the composition of the mixtures or by surface modification or treatment. The higher shear strength was achieved only by increasing the normal load.

## 4. Conclusions and Further Outlook

This research demonstrated the advantage of using button foil to create protrusions in concrete structures. These protrusions increased the shear strength at the UHPC–NSC interface almost to the value of the NSC shear strength. Button foil can be also used effectively for large-scale elements such as bridge girders.

A high shear strength of the interface was achieved by appropriate surface treatment even without reinforcement. The shear strength measured on the box-type specimens was very good for both types of material interface. The residual strength after reaching the shear strength of the interface was significantly improved.

The use of secondary binding materials in the filling concrete can lead to a reduction of the carbon footprint. This is a very actual topic due to the emission allowance system. The use of the NSC2 in the samples resulted in the reduction of the peak value of the load-bearing capacity of the interface. The residual strength was almost identical for both types of NSC, and the possibilities of use of this interface are thus almost the same. In the case of box-type specimens, the contact area also had a very good residual strength. Experiments showed that this contact surface modification was applicable on a larger scale to real structures.

This type of composite specimens or beams can also be implemented in a variant where the UHPC is replaced by steel plates. Steel connectors are also needed to achieve the desired coaction [34]. A UHPC–NSC beam will be easier to manufacture, maintenance-free and probably cheaper. The UHPC shell may also contain prestressed reinforcement. The advantage of the steel variant is mainly weight, size and mechanical resistance. Thus, a different option may be preferred in each situation.

The aim of this research was to validate the described technology and the coaction at the interface on small samples. Future work will focus on designing and testing large beam and column specimens. The results of further research will be presented in future articles.

## Figures and Tables

**Figure 1 materials-15-06537-f001:**
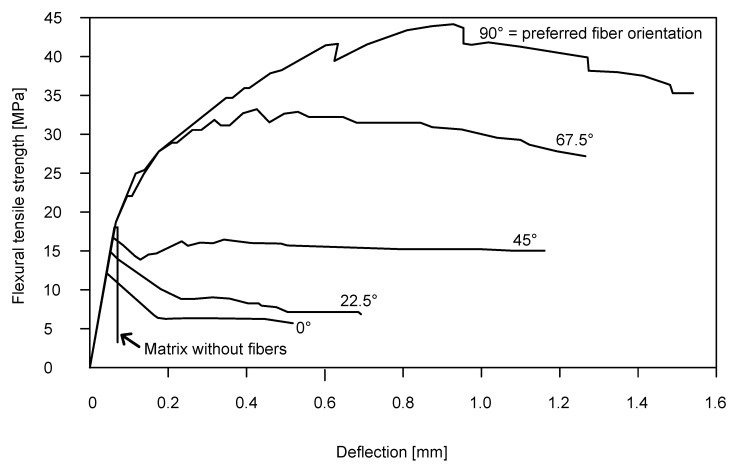
Effect of fibre orientation on the tensile strength of UHPC [1]. The preferred fibre orientation is parallel to the direction of tensile forces, which corresponds to the angle of 90°.

**Figure 2 materials-15-06537-f002:**
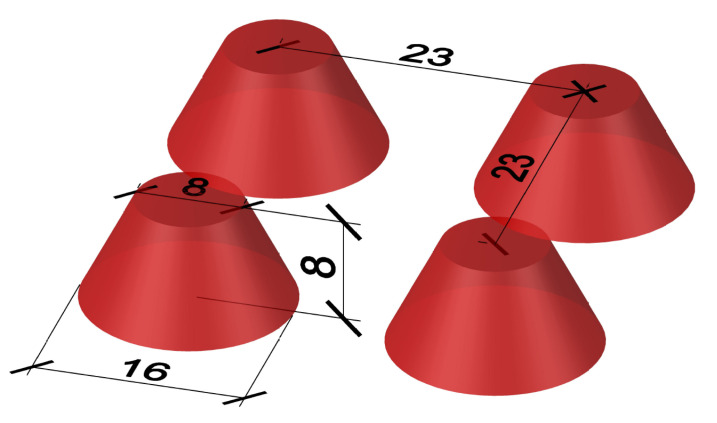
UHPC protrusions made by button foil.

**Figure 3 materials-15-06537-f003:**
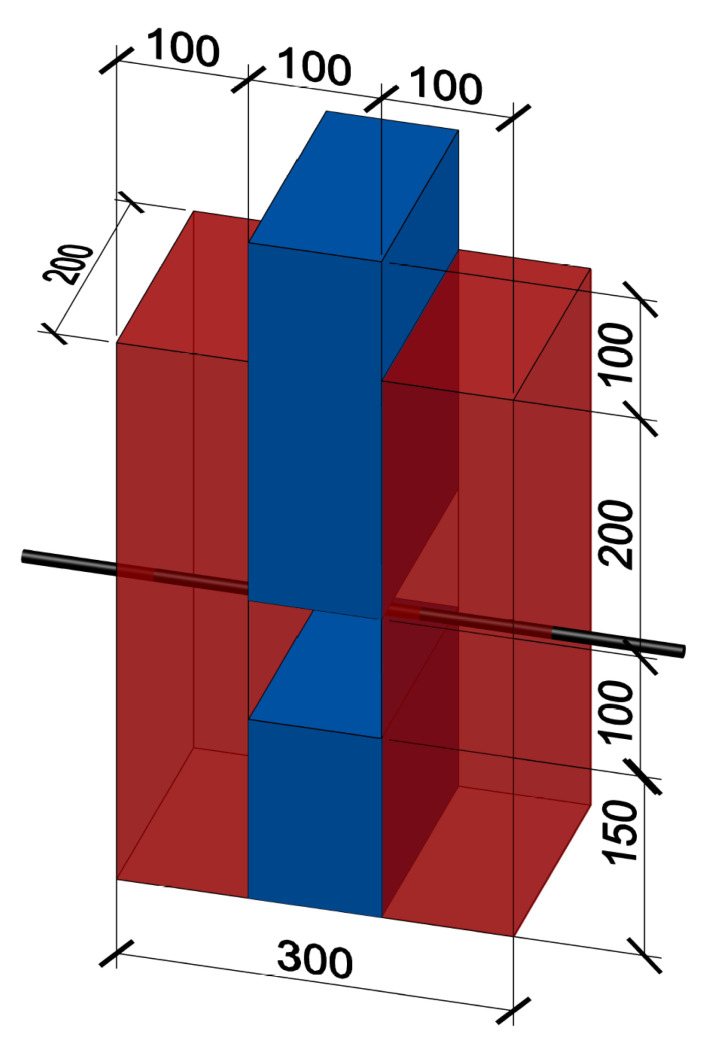
Sandwich specimen, red colour indicates UHPC, blue colour indicates NSC, grey bar is a threaded rod and the empty space between NSC parts is filled with Styrofoam.

**Figure 4 materials-15-06537-f004:**
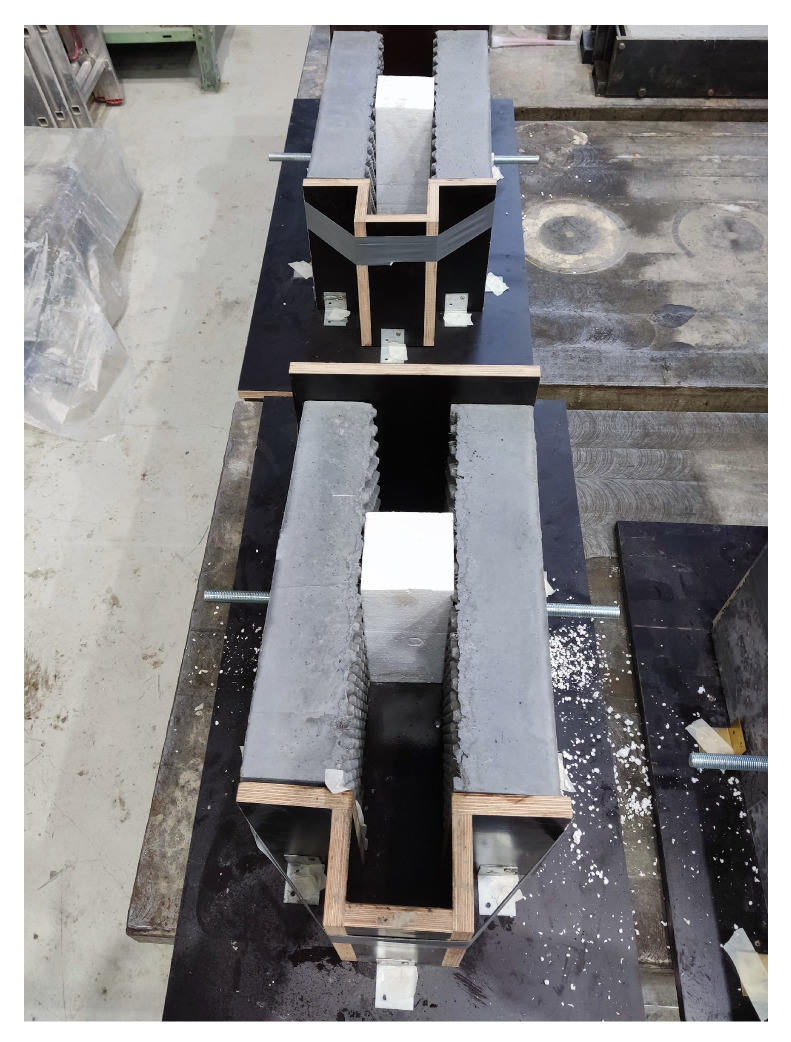
Sandwich samples prepared to be filled with NSC between two UHPC parts.

**Figure 5 materials-15-06537-f005:**
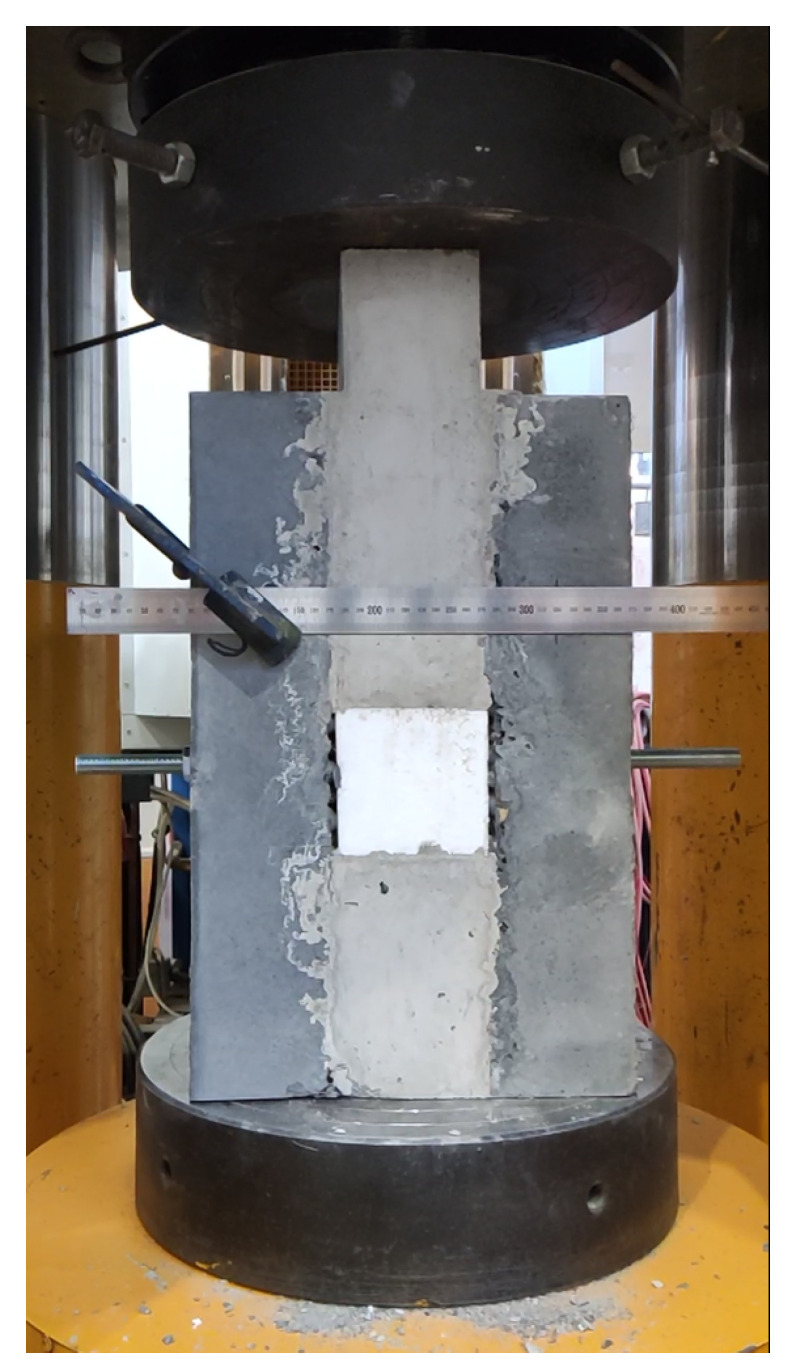
Sandwich sample prepared to be tested in hydraulic loading machine. The gauge is used to determine the formation of the crack.

**Figure 6 materials-15-06537-f006:**
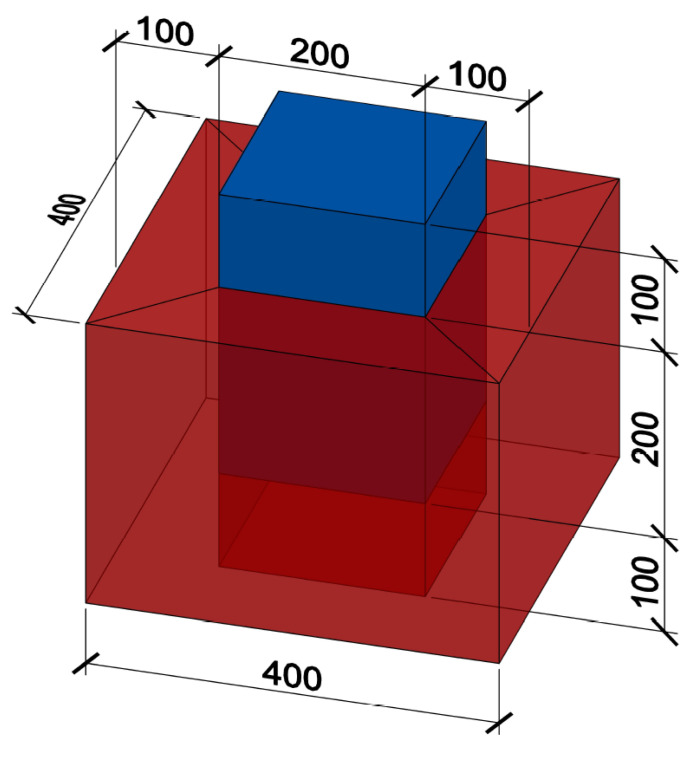
Box specimen, red colour indicates UHPC, blue colour indicates NSC and empty space under the NSC part is filled with Styrofoam.

**Figure 7 materials-15-06537-f007:**
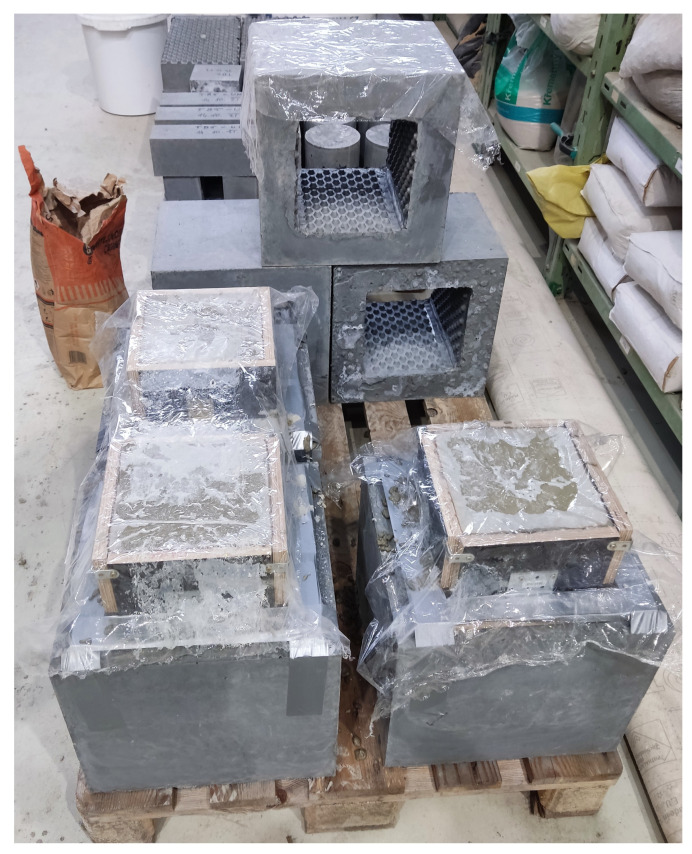
In the front section are box samples for testing of the interaction between UHPC and NSC1; behind them are parts of samples made from UHPC with buttons on the inner surface prepared to be filled with NSC2 with FBC-TB.

**Figure 8 materials-15-06537-f008:**
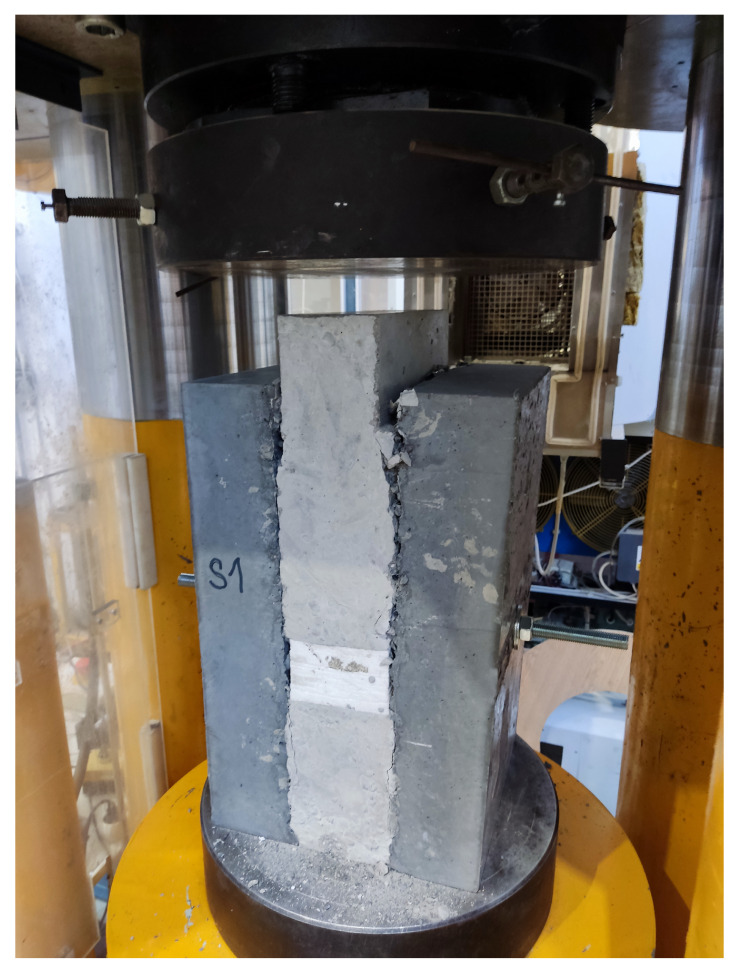
Sandwich sample in the hydraulic loading machine after testing.

**Figure 9 materials-15-06537-f009:**
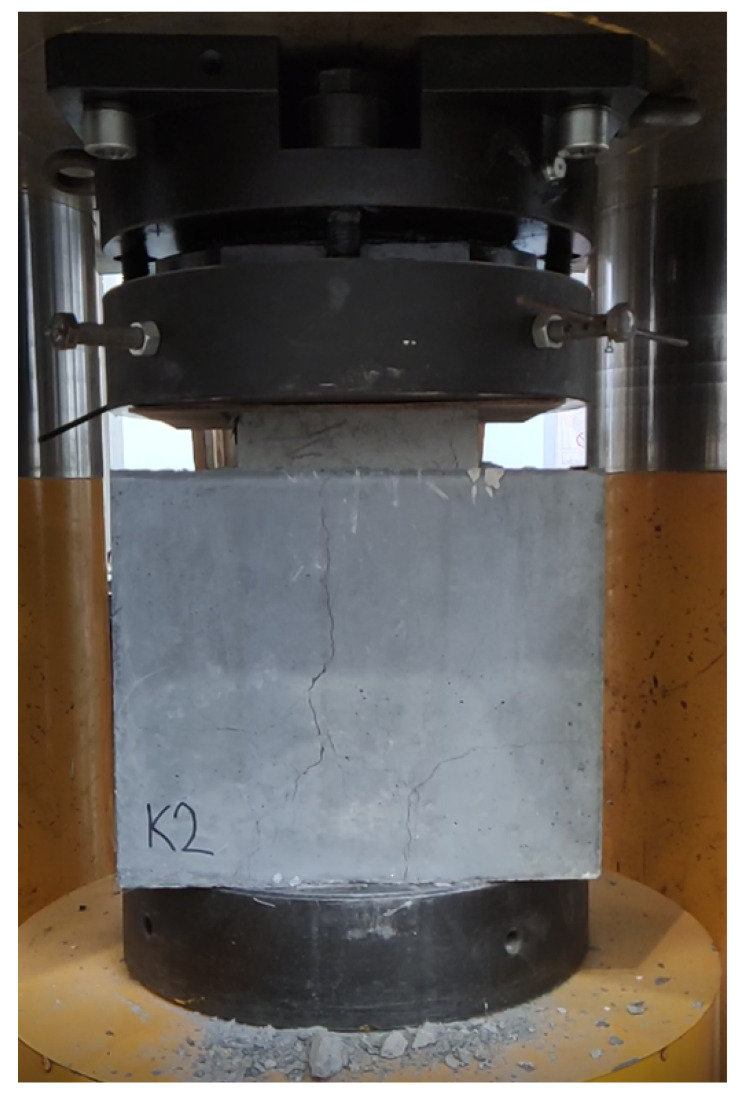
Box sample in the hydraulic loading machine during testing after the crack occurred.

**Figure 10 materials-15-06537-f010:**
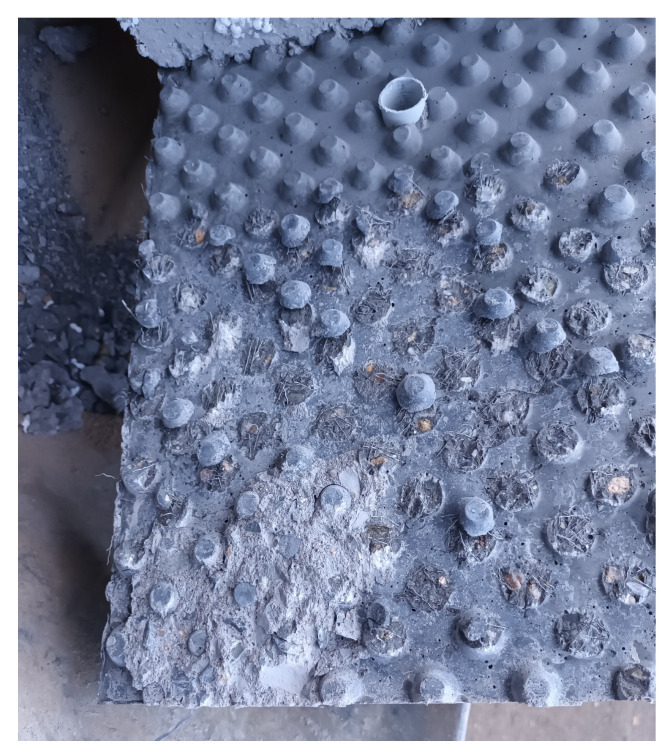
In the specimen, the buttons were sheared off in most of the area, but in some parts crack occurred in the filling concrete.

**Figure 11 materials-15-06537-f011:**
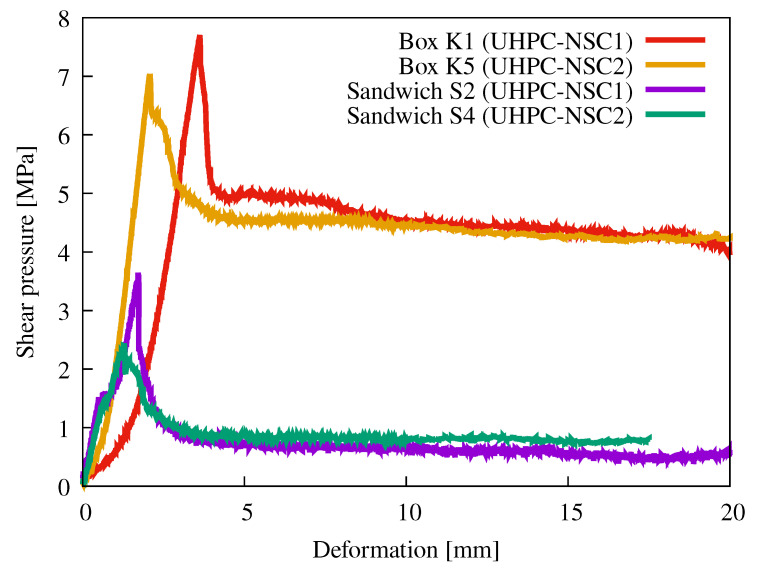
Typical graphs of shear pressure vs. deformation for box and sandwich samples for both types of material interface. The slope of the curves (red and yellow) is affected by a wooden plate placed between the sample and metal surface of the hydraulic loading machine to prevent unwanted damage of the top part of the sample.

**Figure 12 materials-15-06537-f012:**
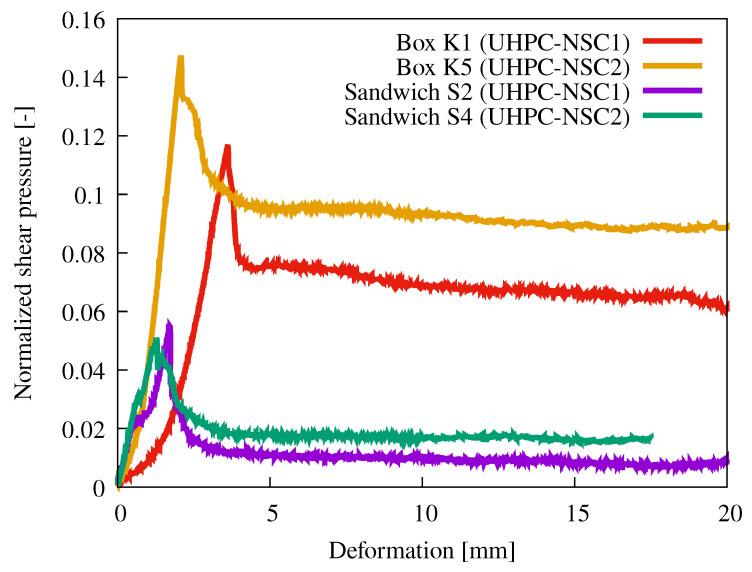
Graphs of normalized shear pressure vs. deformation according to Figure 11. The shear strength was normalized by the average compressive strength measured on cubes, see Table 2.

**Table 1 materials-15-06537-t001:** Composition of both types of NSC.

Component	NSC1	NSC2
	% by Weight	% by Weight
Coarse aggregate	33.0	33.0
Fine aggregate	44.0	44.0
Cement	15.7	7.8
FBC-TB	0.0	7.8
Water	7.2	7.2
Admixtures	0.2	0.2

**Table 2 materials-15-06537-t002:** Material parameters of UHPC and NSC.

		UHPC	NSC 1	NSC 2
C-Cube	1	138	71.4	48.6
(MPa)	2	130	61.7	45.3
	3	152	64.4	49.3
	4	138		
	5	136		
	6	145		
	7	137		
	8	133		
	9	137		
C-Cylinder	1	128	67.3	44.4
(MPa)	2	146	57.1	48.7
	3	91	65.4	45.8
T-Simple	1		4.2	3.2
(MPa)	2		4.0	failed
	3		4.2	3.0
T-Bend-3	1	16.8	8.1	6.6
(MPa)	2	19.4	7.5	6.1
	3	20.5	6.6	6.5
	4		6.5	5.4
	5		6.5	6.4
	6		6.8	6.1
T-Bend-4	1	27.3	8.7	8.0
(MPa)	2	27.6	7.5	7.3
	3	29.0	8.6	7.3
	4	21.9		
	5	27.8		
	6	24.7		
E	1	50.7	36.2	32.1
(GPa)	2	45.6	36.2	33.3
	3	47.5	37.4	32.1

**Table 3 materials-15-06537-t003:** Shear strength of the sandwich samples.

		UHPC–NSC1	UHPC–NSC2
Sandwich	1	4.0	2.4
(MPa)	2	3.7	2.7
	3	failed	2.3

**Table 4 materials-15-06537-t004:** Shear strength of the box samples.

		UHPC–NSC1	UHPC–NSC2
Box	1	7.7	8.6
(MPa)	2	8.5	7.0
	3	8.2	8.0

## Data Availability

The data used to support the findings of this study are available from the corresponding author upon request.

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
