# Peer review of "Experimental Investigation of Cohesion between UHPC and NSC Utilising Interface Protrusions"

_materials, 2022, doi:10.3390/ma15196537_

Round 1

Reviewer 1 Report

The manuscript, entitled "Experimental investigation of cohesion between UHPC and NSC utilizing interface protrusions" is a well written one, presenting the experimental approaches for improving the cohesion between UHPC and NSC. The scope is clear, and the experimental design is sound. However, there are some aspects that the authors could improve before the manuscript gets accepted by MDPI Materials.

Major Points:

1. Introduction: The authors presented a significant amount of content on modeling (Line 79-97). Clearly, numerical modeling is closely related to the experimental work and the motivation of this research, but I would suggest that the authors can shorten these contents in 1.1 "Technical Background". The modeling background does not affect the main context too much. Simplifying this part can help readers to keep focused on the experimental part without being too overwhelmed. 

2. Research Significance. Session 1.2 is well written, presenting the scope of the overall project, but I would recommend adding another paragraphing introducing the content of this specific manuscript (a high level summary). This can help guiding the readers to the main context.

3. The materials information in the manuscript is not very clear. The authors give a detailed description on the sample design, and testing methods, but the materials providers. Please provide the information about the providers of UHPC (and other raw materials).

Minor points:

There are a few unclear sentences that the authors need to edit:

For examples: Line 25-26: " On the other hand there is a risk of brittle fracture not only under extreme load due to the higher strength." The meaning of this sentence is not clear. Do the authors mean "higher loading force" or "higher deformation"?

Line 79-80: "This is why numerical models not only using the finite element method should focus on the best possible approximation of the interface" Changing to "This is why numerical models should focus on the best possible approximation of the interface" seems better to me.

Line 141: "Another used material used in this study" can be "Another material used in this study"

After the revision, I believe this is a very good quality manuscript for Materials.

Reviewer 2 Report

This study provides an experimental investigation of cohesion between UHPC and NSC utilizing interface protrusions. This will increase the contact's tensile and shear strengths, which will increase the composite structure's capacity. This study involved cutting away a portion of the NSC after inserting a button foil into the UHPC formwork. Based on the revision procedure, this article could be accepted after addressing the following comment carefully:

1-      The abstract should be improved. A brief about the performed experimental program should be provided.

2-      The introduction section should be improved. Some studies performed by well-know researchers like Xu and Li could help improve your study. Some of useful studies are provided: (https://doi.org/10.1016/j.cscm.2022.e01321, https://doi.org/10.1016/j.engstruct.2022.114358 , https://doi.org/10.3390/app112110057,  https://doi.org/10.3390/app11209696, https://doi.org/10.1061/(ASCE)ST.1943-541X.0003177, )

3-      The novelty of this study and the gap of previous studies resulted in doing the current one should be discussed in details at the end of abstract section.

4-      The reason for considering two NSC should be discussed in details

5-      The authors referred to Ref [25] for some of tests. But it is necessary to provide the standards that been used.

6-      The number of references is not enough to support the background of this study

7-      This is recommended to replace Figure 1 and its associate in Results and Discussion Section

8-      It is better to normalized the vertical axis of Figure 11 based on the compressive strength.

9-      The conclusion section should be improved and the both quantity and quality analysis should be performed

10-  The obtained results should be compared with previous investigation and other type of panels. For example, https://doi.org/10.3390/ma14185185

Based on those represented above, this paper should be considered for another review after major revision.

Reviewer 3 Report

This paper presents an experimental study examining the shear strength of  a UHPC-NSC interface.

Please consider the following comments:

1.      The paper should be rewritten in a more fluent way

2.      Please add the ingredients of UHPC in a table and include the properties of the steel fibers.

3.      Control specimens are missing. It is important to show the effect of the proposed technique compared to a usually used (with no protrusion) interface

4.      For design purpose, one would use the residual shear strength. Therefore, the advantage of this method should be further verified and explained.

Round 2

Reviewer 1 Report

I appreciate the authors' efforts in revising this manuscript. The quality of the manuscript is significantly improved. I would recommend the acceptance of this work.